# Dynamics of Carbon Storage and Its Drivers in Guangdong Province from 1979 to 2012

**Tao Li** , **Ming-Yang Li \*** and **Lei Tian**

School of Forestry, Nanjing Forestry University, Nanjing 210037, China; litao3014@126.com (T.L.);
tianlei@njfu.edu.cn (L.T.)
* Correspondence: lmy196727@njfu.edu.cn; Tel.: +86-025-8542-7327

**Abstract:** Accurately estimating the carbon storage of forest ecosystems and deriving the driving factors affecting forest carbon storage are the prerequisites and foundations for promoting the development of global carbon sinks. In order to explore an effective approach for monitoring the carbon sink of forests in developed coastal areas on a large scale, in this paper, Guangdong Province was taken as the case study region; eight periods plots of data of national forest resources continuous inventory were used to estimate the forest carbon storage and carbon density in Guangdong Province from 1979 to 2012; unary linear regression and standard deviation ellipse were used to analyze the dynamic change of carbon storage; and the structural equation model was used to study the driving factors of forest carbon storage. The results showed that: (1) From 1979 to 2012, the forest carbon storage in Guangdong Province increased by $15,087.93 \times 10^4$ t, and the forest carbon density increased by 17.66 t/ha. (2) After 2007, the main body of forest carbon storage changed from coniferous species to broadleaf species. (3) From 1979 to 2012, the proportion of young and middle-aged forest carbon storage continued to decline, but it still occupied the dominant component. (4) The forest carbon storage and carbon density in the northern region of Guangdong Province are higher than those in the southern region. (5) Stand factors and environmental factors have a positive effect on forest carbon storage, and understory factors have a negative effect. In conclusion, although forest carbon storage has fluctuated under the influence of forestry policies and human activities, the overall carbon storage and carbon density of Guangdong Province have been increasing. Tree species have become more abundant and the proportion of coniferous forest to broadleaf forest became more rationalized. The forest age group structure is continuously optimized. We also compared our results with that of other provinces in China and other countries with approximate latitude and climatic conditions. The carbon sink potential of Guangdong Province is huge in the future.

**Keywords:** forest carbon storage; carbon density; spatiotemporal dynamics; structural equation model; driving factors; Guangdong province





## 1. Introduction

In order to cope with global climate change, the United Nations formulated the United Nations Framework Convention on climate change in 1992 in order to comprehensively control the emission of $CO_2$ and other greenhouse gases [1,2]. The global carbon cycle has become one of the core issues of global climate change research. As the main body of the terrestrial ecosystem, the forest ecosystem is the largest carbon pool in the terrestrial ecosystem. The fixed carbon by forest every year accounts for about two-thirds of the entire terrestrial system [3–5]. Therefore, forest ecosystem plays an irreplaceable role in keeping the global carbon balance, alleviating the rise of greenhouse gas concentration such as $CO_2$ in the atmosphere and regulating the global climate [6]. The carbon sequestration capacity of forest ecosystems depends on the comparison of carbon input rate and carbon output rate [7]. Forest carbon sequestration capacity, carbon storage and its temporal and spatial dynamic distribution have become hot topics of forest and ecological research in China

and abroad [8]. At present, there are still some uncertainties in accurately estimating the size of the forest ecosystem carbon pool and the carbon flux between the related carbon pools, and the calculation results are often quite different [9,10]. Accurate estimation of the carbon storage of forest ecosystems is a prerequisite for a comprehensive understanding of the status and role of forest ecosystems in the carbon cycle. At this stage, estimation of forest ecosystem biomass is the basis for forest ecosystem carbon storage estimation [11,12], so accurate estimation of forest biomass is the great significance the earth's carbon cycle.

Forest biomass can be converted into carbon storage by the percentage of carbon in the dry weight organic matter of plants (i.e., carbon conversion coefficient) [13]. When estimating carbon storage, the first step is to estimate forest biomass. At present, there are mainly two ways of forest carbon estimation, namely direct measurement and indirect estimation [14]. Direct measurement is conducted by field survey, which is highly accurate. However, this measurement is time-consuming, labor-intensive, and extremely destructive to the ecosystem. Thus, this direct measurement method is usefully adopted at forest stand or ecological scale only. Stand and ecosystem scale usually adopt the direct method. Indirect estimation mainly includes three methods: sampling inventory, model simulation, and remote-sensing based estimation. On large scales, the ground survey data of forest carbon, together with multi-source remote sensing images are collected to estimate landscape regional forest carbon. There are many factors affecting the accuracy of remote sensing estimation, such as variable selection, modeling methods, so the estimation accuracy varies greatly and the prediction results cannot be extended to other regions. Model simulation methods [15], such as climate–vegetation models, biogeographic models, and biogeochemical models, are commonly used to estimation forest carbon at national and global scales [16,17]. The eddy covariance method is a kind of direct measurement of carbon fluxes of micrometeorology [18–20]. This method has the advantages of continuous observation, no interference to the environment and a large observation spatial scope. However, because of the small number of carbon flux observation stations and the influence of terrain and air conditions, its application in country and global carbon storage estimation is limited. Due to the shortcomings of each estimation method, in order to reduce the estimation error, multiple methods are often combined together to estimate the forest carbon in the same research area. At the present stage, the method of sample plot inventory is relatively more accurate, and there are many methods for estimating biomass based on sample plot data. Due to the differences in tree species and regions, the biomass results of different methods often have some errors. For example, Brown [21] and Fang [22] et al. calculated the biomass of different forest types and forests in different countries by biomass expansion factor. Kim [23] compared of allometric equations and biomass expansion factor to calculate the subtropical broadleaf species in South Korea. Park [24] used biomass expansion factor, allometric equation and stand biomass to calculate the biomass of Pinus thunbergii (*Pinus thunbergii* Parl.) in Southern Korea. Biomass expansion factor method can accurately estimate forest biomass. In summary of previous studies [25–31], we selected the commonly recognized biomass expansion factor method in this work.

Guangdong Province is located in the southeastern part of the Eurasian continent and borders the Pacific Ocean to the south. It is severely affected by the monsoon climate. There are many storms and rains in summer. Under this climate condition, forest water conservation, soil and water conservation and other service functions are particularly important. As Guangdong Province is less affected by the Quaternary Ice Age, the flora here has a long history, is rich in forest plant species, preserves many ancient plant species, and forms a flora with ancient plants and relics plants, including 10 endemic genera of plants, accounting for about 5.1% of the endemic genera in China. Guangdong is the gene bank of tropical tree species and animal resources in China and an important part of the southern collective forest area [32–35]. Since the reform and opening up, the process of urbanization and industrialization has been accelerating, the forest has been disturbed and destroyed by human activities for a long time, the proportion of primary forest vegetation types have been decreasing, and the habitat conditions for endangered

species have been deteriorating. Therefore, the forest resources in Guangdong Province are of great significance to the development of forestry carbon sequestration and scientific forestry research in China.

At present, there is only a small amount of carbon sink function studies based on the national forest resources continuous inventory data (NFCI) of forest resources at provincial scales. In these studies, the biomass conversion factor method was always used to carry out the dynamic analysis of carbon storage and carbon density of different forest types in time dimension, lacking the analysis of spatial distribution, spatial trend changes, and driving factors. At present, there are no relevant reports on the analysis of spatial distribution and spatial change of forest carbon storage based on standard deviation ellipse (SDE) and the analysis of driving factors of forest carbon density using structural equation model (SEM). The main objectives of this paper are as follows: (1) to explore an applicable method to estimate the forest carbon storage on large spatial scale using NFCI plots data; (2) to reveal the spatiotemporal trend of forest carbon in typical rapidly urbanizing province of China; and (3) to identify driving factors of forest carbon storage to provide a scientific basis for making sustainable forest management plan.

## 2. Materials and Methods

### 2.1. Study Area

Guangdong Province is located in the southernmost part of mainland China (Figure 1). The whole territory is bounded by latitude 20°09′~25°31′ N, longitude 109°45′~117°20′ E. The total land area is 179,800 km$^2$, accounting for approximately 1.87% of the country's land area. The landforms types of Guangdong Province are complex and diverse, including mountains, hills, and plains, which account for 33.7%, 24.9%, and 21.7% of the total land area of the province, respectively. Rivers and lakes only account for 5.50% of the total land area of the province. The provincial terrain is generally high in the north and low in the south, with mountains and high hills in the north. Guangdong Province belongs to the East Asian monsoon climate. From north to south, there are central subtropical, southern subtropical, and tropical climates. It is a province rich in sunshine, heat, and water resources in China. The precipitation is mainly concentrated from April to September, with an annual average temperature of 21.9 °C and an annual average precipitation of 1790 mm. Affected by climatic conditions, there are a wide range of vegetation and vegetation communities in Guangdong with banded distribution. From south to north, there are tropical seasonal rain forest, subtropical monsoon evergreen broadleaf forest, and typical evergreen broadleaf forest in middle subtropics. By 2020, the forest area of Guangdong Province was 10,524,100 hm$^2$ [36]. There are many kinds of animals and plants in Guangdong. The national first-level protected plants include three species of cyathea (*Alsophila spinulosa* (Wall. ex Hook.)), cathaya (*Cathaya argyrophylla* Chun et Kuang), and tigridiopalma magnifica (*Tigridiopalma magnifica* C. Chen). The national second-level protection includes 24 plants species such as metasequoia (*Metasequoia glyptostroboides* Hu and W. C. Cheng) and Whitearilyew (*Pseudotaxus chienii* (Cheng) Cheng).

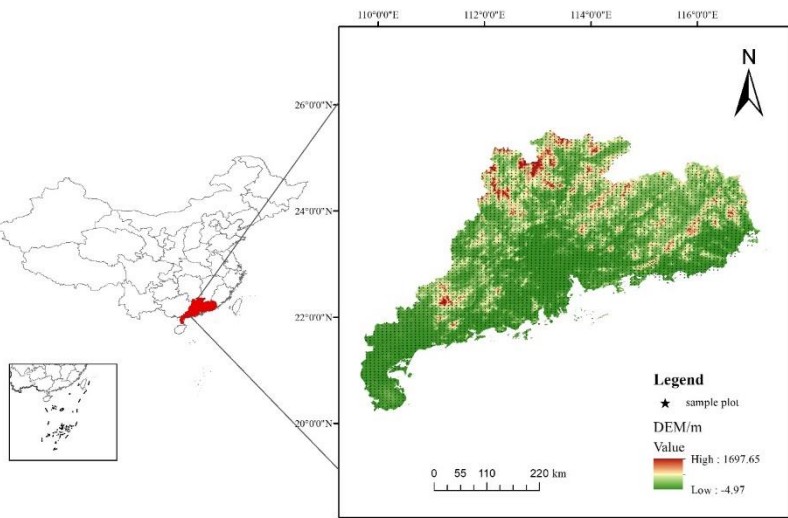

**Figure 1.** Location of Guangdong Province and sample plots.

*2.2. Data Acquisition and Preprocessing*

This paper estimates the carbon storage of Guangdong Province based on the sample plots data of 8 consecutive the NFCI in Guangdong Province from 1979 to 2012. The NFCI takes provinces as the sampling population and adopts systematic sampling. According to the actual situation of each province, the sampling interval of each province is determined on the kilometer grid. Permanent sample plots are set up to conduct forest resource surveys. Guangdong Province conducts systematic sampling based on 6 km × 8 km spacing, with a total of 3685 fixed sampling plots with an area of 0.067 hm² each. The attributes of these plots include slope, slope direction, slope position, altitude, soil name, soil layer thickness, soil texture, humus thickness, average age, average diameter at breast height (DBH), average tree height, canopy density, tree species structure, live tree stock, and other investigation factors.

Before the estimation of forest carbon storage and carbon density, the non-forestland sample plots with a stock volume of 0, such as water bodies and buildings, should be deleted from the NFCI. Referring other research methods in China and abroad, the biomass conversion factor method is adopted to convert the forest stock volume of fixed sample plots into forest biomass. Different tree species had different carbon content rates, so biomass was converted into carbon storage based on the carbon content rate of dominant tree species.

*2.3. Research Method*

2.3.1. Estimation of Carbon Storage and Carbon Density

In this paper, the biomass expansion factor equation of each tree species provided by the Guangdong Forest Resources Monitoring Center [37] is used to estimate the forest biomass. For tree species without corresponding equations, the biomass calculation method in the article by Yu C [38] is used.

The calculation formula of arbor is:

$$W_{stem} = a_1 \times D^{b_1} \times H^{c_1} \times V \tag{1}$$

$$W_{branch} = a_2 \times D^{b_2} \times H^{c_2} \times V \tag{2}$$

$$W_{leaves} = a_3 \times D^{b_3} \times H^{c_3} \times V \tag{3}$$

$$W_{roots} = a_4 \times D^{b_4} \times H^{c_4} \times V \tag{4}$$

$$W = W_{stem} + W_{branch} + W_{leaves} + W_{roots} \tag{5}$$

where $W_{stem}$ is biomass of tree stems in t/ha; $W_{branch}$ is biomass of tree branches in t/ha; $W_{leaves}$ is biomass of tree leaves in t/ha; $W_{roots}$ is biomass of tree roots in t/ha; $W$ is the total biomass in t/ha; $D$ is average DBH in cm; $H$ is average height in m; $V$ is forest stock in m/ha; and $a_i$, $b_i$, and $c_i$ are coefficients. See reference [37,38] for the calculation formula of biomass of different forest types. Bamboo biomass is calculated according to 22.5 kg per plant [11].

Forest carbon storage can be calculated by multiplying forest biomass by carbon content, and the calculation formula is:

$$C_\rho = W \times C_c \tag{6}$$

$$C_0 = C_\rho \times S \tag{7}$$

where $C_\rho$ is the carbon density in t/ha that is the carbon storage per unit area; $W$ is the biomass per unit area in t/ha; $C_c$ is the carbon content coefficient with no unit (Table 1); $C_0$ is the total forest carbon storage in t; and $S$ is the forest area in hm.

**Table 1.** Carbon content coefficient of different tree species (excerpt) [39].

| | Carbon Content Coefficient |
|---|---|
| Pinus massoniana | 0.544 |
| Cunninghamia lanceolata | 0.555 |
| Hard broadleaved forest | 0.522 |
| Soft broadleaved forest | 0.520 |
| Eucalyptus | 0.521 |

### 2.3.2. Slope Univariate Linear Regression Analysis

The univariate linear regression equation can be used to analyze the change trend of carbon storage in the study area with time, and its calculation formula is [40]:

$$\theta_{slope} = \frac{n \times \sum_{i=1}^{n} i \times C_i - \sum_{i=1}^{n} i \sum_{i=1}^{n} C_i}{n \times \sum_{i=1}^{n} i^2 - \left(\sum_{i=1}^{n} i\right)^2} \tag{8}$$

where $\theta_{slope}$ is trend slope; $n$ is the number of study periods ($n = 8$); and $C_i$ is the forest carbon storage in the $i$-th year. If $\theta_{slope}$ is positive, it indicates that the change of forest carbon storage is increasing year by year. If $\theta_{slope}$ is zero, indicating that forest carbon storage is basically stable during the study period. If $\theta_{slope}$ is negative, it indicates that forest carbon storage is decreases during the study period.

### 2.3.3. Standard Deviational Ellipse

Standard deviational ellipse (SDE) [41–44] is an analysis method to characterize the spatial distribution characteristics, SDE can accurately reveal the various characteristics of the spatial distribution pattern of geographical elements that include four basic elements: the center of gravity coordinate, the rotation angle, and the standard deviation along the long axis (i.e., Y axis) and the short axis (i.e., X axis). These elements, respectively, represent the relative position of the spatial distribution pattern of elements, the main trend direction of development, and the degree of dispersion in the main and secondary directions. The size of the ellipse reflects the concentration of the overall elements of the spatial pattern. The Y axis of the ellipse indicates the direction of data distribution, and the X axis indicates the range of data distribution. The shorter the X axis, the more obvious the spatial aggregation of the data; the longer the X axis, the greater the degree of dispersion of the data. That is, the oblateness indicates the degree of clarity of the direction of the data and the degree of centripetal force. In this paper, ArcGIS 10.3 was used to generate the standard deviation ellipse of carbon storage in Guangdong Province to identify the position of the center of gravity and the spatial movement trend of carbon storage from 1979 to 2012.

Mean center coordinate:

$$\overline{X}_w = \sum_{i-1}^{n} w_i x_i / \sum_{i=1}^{n} w_i \tag{9}$$

$$\overline{Y}_w = \sum_{i-1}^{n} w_i y_i / \sum_{i=1}^{n} w_i \tag{10}$$

Azimuth:

$$\theta = \arctan\left[\left(\sum_{i=1}^{n} x_i'^2 - \sum_{i=1}^{n} y_i'^2\right) + \sqrt{\left(\sum_{i=1}^{n} x_i'^2 - \sum_{i=1}^{n} y_i'^2\right)^2 + 4\left(\sum_{i=1}^{n} x_i' y_i'\right)^2}\right] / 2 \sum_{i=1}^{n} x_i' y_i' \tag{11}$$

Axis standard deviation:

$$\delta_x = \sqrt{\sum_{i=1}^{n} \left(x_i' \cos\theta - y_i' \sin\theta\right)^2 / n} \tag{12}$$

$$\delta_y = \sqrt{\sum_{i=1}^{n} \left(x_i' \sin\theta - y_i' \cos\theta\right)^2 / n} \tag{13}$$

where $\left(\overline{X_w}, \overline{Y_w}\right)$ is the weighted average center coordinate; $(x_i, y_i)$ is the spatial position coordinate of each element; $w_i$ is the weight; $\theta$ is the ellipse azimuth angle; $(x', y')$ is the relative coordinate of each point from the center of the area; and $\delta_x$ and $\delta_y$ are the standard deviation along the x-axis and y-axis, respectively.

2.3.4. Structural Equation Model

Structural equation model (SEM) [45–49] is an advanced and robust multivariate statistical method that combines factor analysis and regression analysis, allowing hypothesis testing on a complex network of path relationships to analyze the relationship between measured variables and latent variables, as well as the relationship between each latent variable. SEM is composed of measurement model and structural model. The former is used to analyze the relationship between measurement variables and latent variables, and the latter is used to analyze the relationship between latent variables.

Measurement model:

$$X = A_x \xi + \delta \tag{14}$$

$$Y = A_y \eta + \varepsilon \tag{15}$$

Structural model:

$$\eta = B\eta + \Gamma\xi + \zeta \tag{16}$$

where $X$ represents the exogenous measurement variable, $Y$ represents the endogenous measurement variable; $\xi$ represents the exogenous latent variable, $\eta$ represents the endogenous latent variable; and $A_x$ represents the factor loading matrix of $X$ on $\xi$. That is, the coefficient matrix reflecting the strength of the relationship between the exogenous index and the exogenous latent variable. $A_y$ is the factor loading matrix of $Y$ on $\eta$, that is, the coefficient matrix reflecting the strength of the relationship between endogenous indexes and endogenous latent variables. $\delta$ represents the measurement error of exogenous measurement variables, and $\varepsilon$ represents the measurement error of endogenous measurement variables. $B$ represents the structure coefficient matrix of the relationship between endogenous latent variables; $\Gamma$ represents the structure coefficient matrix of the relationship between endogenous latent variables and exogenous latent variables; and $\zeta$ represents the disturbance or residual in the structural model.

SEM can study not only observable variables, but also the relationship of variables that cannot be observed directly. It can study not only the direct effect between variables, but also the indirect effect between variables. It can handle multiple dependent variables

at the same time. It allows independent variables and dependent variables to contain measurement errors. It allows independent variables and dependent variables to contain measurement errors. The relationship between variables can be visually displayed through the path diagram. Researchers can construct the relationship between implicit variables and verify whether this structural relationship is reasonable. It can decompose the correlation coefficient to investigate the direct and indirect effects of one variable on another [50,51]. SEM also has some limitations [52]. The main problem affecting the interpretation ability of SEM is the specified error, but the SEM program cannot test the specified error at present. At the same time, SEM has high requirements for sample size, and it also requires that the model must meet the recognition conditions, and it cannot deal with the real classification variables. SEM can be evaluated from many aspects, such as model regression coefficient, load coefficient, and model fitting index. The regression coefficient of the model shows the influence relationship between the latent variables through the non-normalized and normalized path coefficients. The load factor is that when the *p* values show a significant level and the standardized factor load factor is greater than 0.5, it indicates that the model measurement relationship is good. If a path does not show a significant relationship ($p > 0.05$), or the load factor is too low, it can be considered to delete this factor from the model. There are many fitting indexes for SEM. In this paper, chi-square degrees of freedom ratio ($\chi^2/df$) are adopted. Comparative fit index (CFI) and root-mean-square error of approximation (RMSEA) were used to evaluate the model [53]. The chi-square degree of freedom ratio is chi-square divided by the value of the degree of freedom, generally between 1 and 3, indicating that the model fits well. CFI is obtained during the comparison between the hypothetical model and the independent model, and its value is between 0 and 1. The closer it is to 0, the worse the fitting; and the closer it is to 1, the better the fitting. Generally, when CFI is greater than 0.9, the fitting of the model is considered to be better. RMSEA is the index of the evaluation model which is fitted, and the range is between 0 and 1. If it is close to 0, it means that the fitting is good; the closer to 1, it means that the fitting condition of the model is worse; if RMSEA = 0 it indicates that the model is fully fitted. When RMSEA is less than 0.05, the model is close to the fully fitting.

## 3. Results

### 3.1. Temporal and Spatial Dynamics of Carbon Storage

3.1.1. Temporal Changes of Forest Carbon Storage by Forest Type and Age Group

According to the estimation method mentioned above, the carbon storage and carbon density of Guangdong Province from 1979 to 2012 years after harvest were calculated (Tables 2 and 3). It can be seen from Table 2 that from 1979 to 2012, the carbon storage and carbon density of Guangdong Province decreased slightly only in 1983, and showed an upward trend in other years. The carbon storage in Guangdong Province was $4372.91 \times 10^4$ t in 1979 and increased to $19,460.84 \times 10^4$ t in 2012. The total increase in 33 years was $15,087.93 \times 10^4$ t, and the average annual increase was $457.21 \times 10^4$ t. According to the forest type, it is divided into four types, arbor forest, bamboo forest, economic forest, and shrub. In each inventory year from 1979 to 2012, the carbon storage of arbor forests accounted for 95.86% (1979), 97.93% (1983), 92.95% (1988), 95.05% (1992), 83.13% (1997), 86.42% (2002), 92.27% (2007), and 93.15% (2012) of the total carbon storage of Guangdong Province, respectively. Bamboo forest accounts for 0.56% (1979), 0.72% (1983), 2.89% (1988), 0.96% (1992), 5.64% (1997), 9.47% (2002), 6.52% (2007), and 5.63% (2012). The percentage of economic forest is 3.58% (1979), 1.35% (1983), 4.16% (1988), 3.96% (1992), 11.24% (1997), 4.10% (2002), 1.17% (2007), and 1.11% (2012). Shrub accounted for 0.03% and 0.11%, respectively, in 2007 and 2012, and there was no record of shrub species in other years. Among arbor forests, the largest increase in carbon storage is in the broadleaf mixed forest. In 1979, the carbon storage of broadleaf mixed forest accounted for 14.48% of the total carbon storage, and in 2012, the carbon storage of broadleaf mixed forest accounted for 37.00% of the total carbon storage, and the average annual increase of carbon storage was $196.84 \times 10^4$ t. The second largest increase of carbon storage is broadleaf forest. In 1979, the

carbon storage of broadleaf forest accounted for 4.08% of the total carbon storage, increased to 22.24% in 2012, and the average annual increase of carbon storage was $125.78 \times 10^4$ t. Carbon storage of coniferous and broadleaf mixed forest increased by $32.71 \times 10^4$ t per year. The average annual growth rate of carbon storage in coniferous forest was $47.40 \times 10^4$ t, but the percentage of carbon storage in total carbon storage showed a downward trend, which decreased from 57.80% in 1979 to 21.02% in 2012, with a total decrease of 36.77%. The average annual increase of carbon storage in coniferous mixed forest was the lowest ($19.55 \times 10^4$ t). It can be seen from the above data that the percentage of carbon storage of broadleaf forest and the total carbon storage have increased steadily. In 2012, the contribution of broadleaf species has exceeded that of coniferous species to the total carbon storage in Guangdong Province. It is mainly due to the construction of ecological public welfare forest and forest stand transformation project in Guangdong Province. The structure of tree species has been adjusted, the planting area of coniferous species such as Chinese fir (*Cunninghamia lanceolata* (Lamb.) Hook.) and masson pine (*Pinus massoniana* Lamb.) has been reduced, while the planting of broadleaf species represented by eucalyptus (*Eucalyptus robusta* Smith) has been strengthened, which has increased the planting area of broadleaf species, improved the forest quality, and steadily increased the forest carbon storage in Guangdong Province.

**Table 2.** Carbon storage in Guangdong Province by forest type.

| | | Carbon Storage/$\times 10^4$ t | | | | | | | |
| | | 1979 | 1983 | 1988 | 1992 | 1997 | 2002 | 2007 | 2012 |
|---|---|---|---|---|---|---|---|---|---|
| arbor | coniferous forest | 2527.36 | 1995.74 | 3151.39 | 4498.73 | 3137.32 | 4184.37 | 3631.22 | 4091.49 |
| | broadleaf forest | 178.31 | 262.85 | 1610.20 | 2178.37 | 4011.03 | 6080.88 | 7093.22 | 4328.95 |
| | coniferous mixed forest | 260.52 | 214.38 | 299.62 | 455.29 | 460.29 | 499.55 | 565.66 | 905.72 |
| | broadleaf mixed forest | 633.38 | 611.16 | 614.60 | 732.34 | 1120.71 | 1175.29 | 2312.27 | 7128.96 |
| | coniferous and broadleaf mixed forest | 592.35 | 570.73 | 777.06 | 1086.41 | 1234.45 | 1605.78 | 1582.78 | 1671.80 |
| bamboo forest | | 24.51 | 26.69 | 200.74 | 90.30 | 675.82 | 1484.36 | 1074.56 | 1095.26 |
| economic forest | | 156.47 | 50.20 | 288.71 | 372.66 | 1346.57 | 643.35 | 192.99 | 216.77 |
| shrub | | | | | | | | 4.83 | 21.87 |
| total | | 4372.91 | 3731.75 | 6942.32 | 9414.10 | 11,986.20 | 15,673.58 | 16,457.53 | 19,460.84 |

**Table 3.** Carbon density in Guangdong Province.

| | | Carbon Density/(t/ha) | | | | | | | |
| | | 1979 | 1983 | 1988 | 1992 | 1997 | 2002 | 2007 | 2012 |
|---|---|---|---|---|---|---|---|---|---|
| arbor | coniferous forest | 9.04 | 10.05 | 12.02 | 12.26 | 11.09 | 16.94 | 21.42 | 21.37 |
| | broadleaf forest | 15.09 | 12.15 | 18.35 | 19.60 | 22.21 | 28.65 | 22.51 | 22.04 |
| | coniferous mixed forest | 11.94 | 8.09 | 12.32 | 14.78 | 18.57 | 22.68 | 24.46 | 25.91 |
| | broadleaf mixed forest | 28.2 | 20.09 | 30.78 | 31.44 | 32.14 | 34.43 | 36.09 | 40.17 |
| | coniferous and broadleaf mixed forest | 12.23 | 6.21 | 16.24 | 16.52 | 17.93 | 22.52 | 24.11 | 25.78 |
| bamboo forest | | 4.05 | 8.47 | 9.99 | 7.78 | 20.38 | 27.83 | 26.53 | 30.31 |
| economic forest | | 4.33 | 11.73 | 7.71 | 6.93 | 6.92 | 8.21 | 7.91 | 9.28 |
| shrub | | | | | | | | 0.49 | 2.98 |
| average | | 12.28 | 11.61 | 14.37 | 14.45 | 16.45 | 22.52 | 23.02 | 26.77 |

It can be seen from Table 3 that the average carbon density of each inventory year in Guangdong Province from 1979 to 2012 was 12.28 t/ha (1979), 11.61 t/ha (1983), 14.37 t/ha

(1988), 14.45 t/ha (1992), 16.45 t/ha (1997), 22.52 t/ha (2002), 23.02 t/ha (2007), and 26.77 t/ha (2012), respectively, with an annual average increase of 0.44 t/ha. The increment of coniferous mixed forest was the highest, with a total growth of 13.97 t/ha. The second is coniferous and broadleaf mixed forests, with a total increment of 13.55 t/ha. The total increment of coniferous forest was 12.33 t/ha. The total increment of broadleaf mixed forest was 11.97 t/ha. The total increment of broadleaf forest is the lowest, which is 6.95 t/ha. The total increment of carbon density of coniferous mixed forest and coniferous forest is higher than that of broadleaf mixed forest and broadleaf forest. This is because coniferous species such as Chinese fir and masson pine are mainly fast-growing forest plantation, so they increase rapidly. Broadleaf species are mainly natural secondary forest and grow slowly. The carbon density of broadleaf mixed forest is the highest and the proportion of mature and over-mature forest is larger, followed by broadleaf forest, coniferous and broadleaf mixed forest, coniferous mixed forest, and coniferous forest is the lowest. Coniferous forests in Guangdong are dominated by artificial secondary forests with a large proportion of middle young forest, so their carbon density is significantly lower than that of broadleaf forest. Moreover, benefiting from the policy of actively building mixed forests of native broadleaf species in Guangdong Province; therefore, the carbon density of broadleaf mixed forest is the highest.

The carbon storage of arbor forest in Guangdong Province is divided into different age groups for analysis, and the percentage of each age group to the total carbon storage is analyzed (Figure 2). From 1979 to 2012, the carbon storage of young forests accounted for 22.96%~42.57% of the total forest carbon storage in Guangdong Province, 39.63% ~ 69.29% was from middle-aged forests, 0.66%~20.53% was from near-mature forests, 1.09%~12.41% was from mature forests, and 0.01%~2.69% was from overmature forests. The carbon storage of middle-aged forests accounted for the largest percentage of total forest carbon storage, followed by young forest, middle-aged forest, mature forest, and overmature forest. The total percentage of young and middle-aged forest reached the highest value (97.57%) in 1979 and then decreased slowly. By 2012, the percentage of young and middle-aged forests reached a value of 64.37%. It can be seen from Figure 2 that the carbon storage in Guangdong Province is mainly distributed in young and middle-aged forests, but the percentage of mature and overmature forests has increased in recent years, and the percentage of age groups tends to be rationalized. Taking the carbon density in 2012 as an example, counted by age group from high to low, the percentage of overmature forest is 47.62%, mature forest is 38.08%, near-mature forest is 35.43%, middle-aged forest is 32.39%, and young forest is 15.66%, respectively. It can be seen that there is a positive correlation between carbon density and tree age, and the carbon density of mature and overmature forest is much greater than that of young forest. Therefore, Guangdong Province should increase the percentage of mature and overmature forest in the future forest management by extending the forest management rotation.

3.1.2. Spatial Dynamics of the Forest Carbon Storage in Guangdong Province

Based on the spline function interpolation theory of ordinary thin disk and local thin disk, ANUSPLIN [54,55] allows the introduction of covariates (such as elevation) in addition to independent variables, and the interpolation result is very smooth and has a strong transition. There is close relationship between forest carbon storage and altitude in Guangdong Province, the carbon storage in northern mountainous areas with higher altitude is generally higher and that in southern coastal plain areas with lower altitude is always lower. Therefore, this paper used ANUSPLIN 4.37 interpolation software to interpolate the carbon density of Guangdong Province by using elevation as a covariate. The interpolation results in 2012 was taken as an example to analyze the regional distribution of carbon storage in Guangdong Province (Figure 3). The carbon density in northern Guangdong Province is higher, such as Shaoguan, Heyuan, Meizhou, Qingyuan, Yunfu, and other prefectures. These northern prefectures are mainly mountainous with many natural mixed forest and less human disturbance. Therefore, the carbon density in this

part of the province is significantly higher than that in other prefectures. In the Pearl River Delta, represented by Shenzhen, Dongguan, and Zhongshan, due to the needs of economic development, the built-up land percentage is higher than that in northern Guangdong, and the forest area is less. Moreover, the new afforestation is mainly artificial forest, and the area ratio of young forest is significant. Therefore, the forest carbon density in this area is significantly lower than the average value of Guangdong Province.

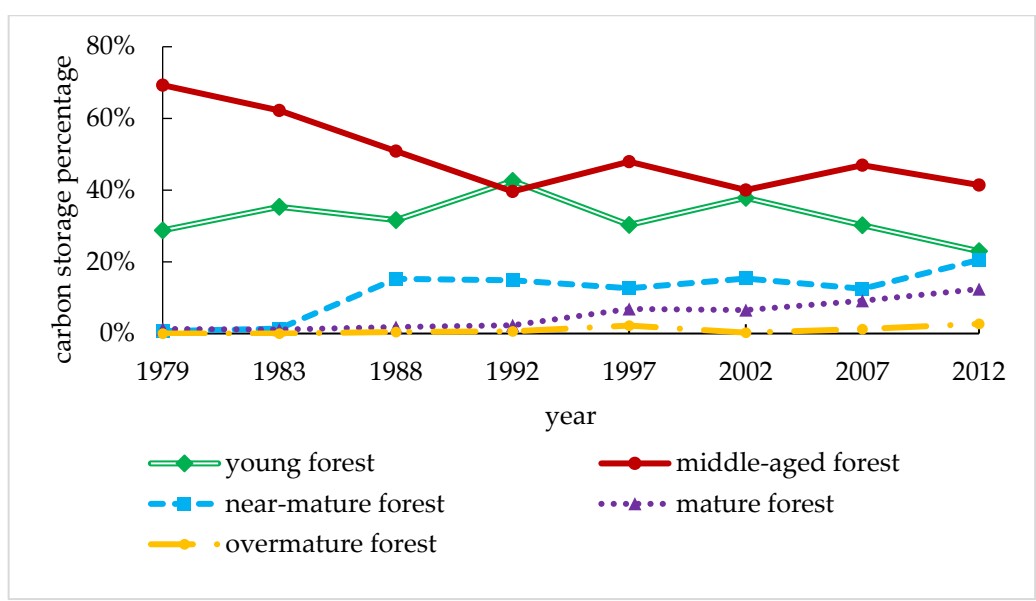

**Figure 2.** The changes of carbon storage percentage on age class of arbor forest.

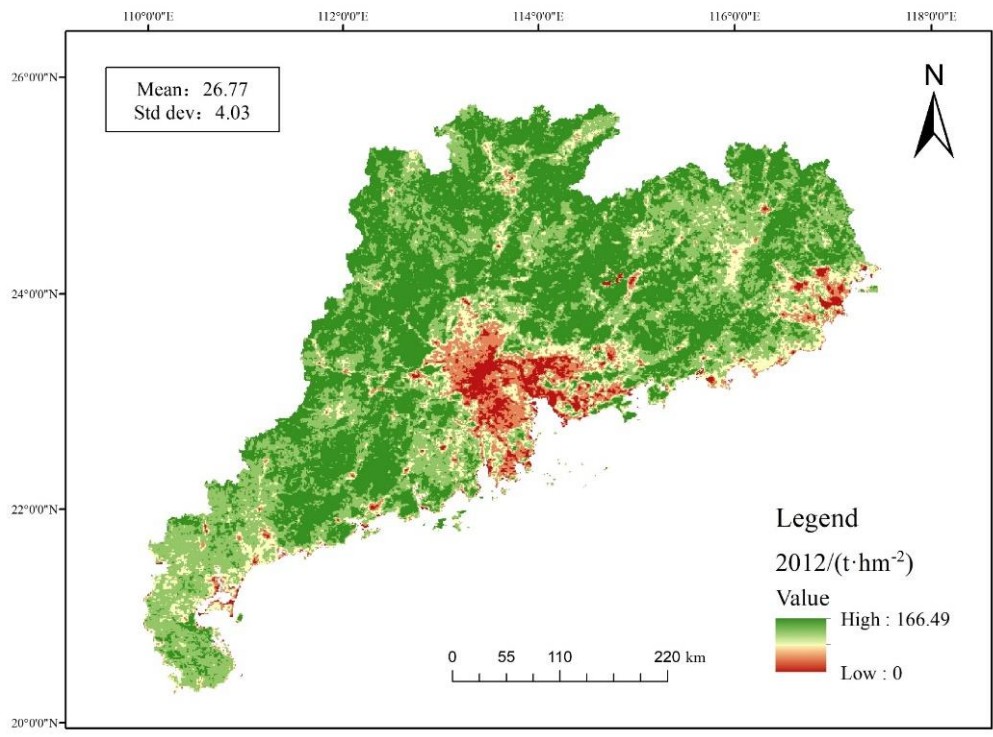

**Figure 3.** Carbon density of Guangdong Province from 1979 to 2012.

Formula 8 was used to obtain the slope of temporal variation trend of forest carbon storage in Guangdong Province from 1979 to 2012. The results showed that $\theta_{slope}$ were all

greater than or equal to 0, indicating that carbon storage in Guangdong Province showed an overall upward trend from 1979 to 2012. The mean value of $\theta_{slope}$ is 3.01, and the standard deviation is 0.5. In order to further understand the temporal change of carbon storage in each prefecture in the study region, four grades are obtained by adding and subtracting 1 times standard deviation of the average value. At the same time, those with a slope of 0 are divided into one grade, and finally five trend slope grades are obtained (Figure 4). The results showed that the area proportions of the five grades of carbon storage in Guangdong Province are basically stable (2.95%), low growth (6.21%), medium growth at 12.38%, higher growth (34.48%), high growth (43.98%). Seen from Figure 4, carbon storage in Guangdong has been increasing at higher levels. It can be seen from Figure 4, the growth of carbon storage in the central and northern regions is higher than that in the southern regions. This is mainly because the Pearl River Delta region is greatly affected by economic and human impacts. During 1979–2012, carbon storage declined twice, which affected the overall increase level. The growth of forest carbon storage in western Guangdong is in the medium and low level, mainly because the development of agriculture, mineral and marine industries in this region has limited the growth of forest area, and the growth of forest carbon storage is relatively slow.

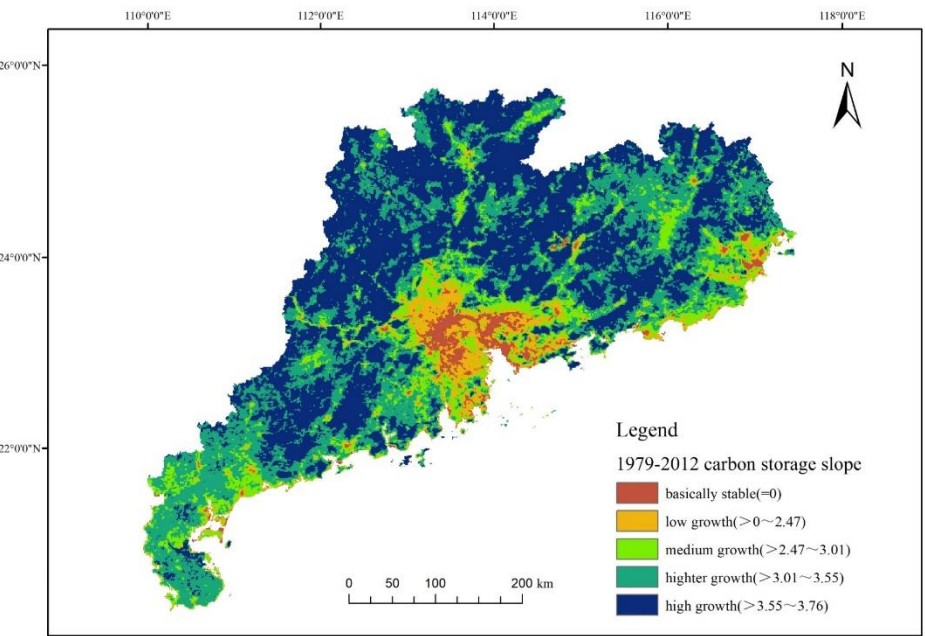

**Figure 4.** Temporal changes of carbon storage in Guangdong Province from 1979 to 2012.

Figure 5 shows the ellipse and center of gravity of the standard deviation of carbon storage in Guangdong Province from 1979 to 2012. From 1979 to 1988, the X axis of the standard deviation ellipse became smaller, the Y axis increased and the oblateness decreased, indicating that the spatial aggregation of forest carbon storage increased and the carbon storage distributed in Guangdong Province was no longer uniform, and the center of gravity shifted to the middle east of Guangdong Province. This is because the implementation of the Three Determinations of Forestry policy after 1979 led to large-scale deforestation, which caused serious damage to the forest resources in the low-altitude areas in the study area, while the forest resources in the high-altitude areas have been protected due to terrain and transportation conditions, so the forest carbon storage are gradually concentrated and the center of gravity is shifted. The oblateness of the standard deviation ellipse increased from 1988 to 1992, decreased from 1992 to 1997, gradually increased from 1997 to 2012, and the center of gravity shifted from the middle to the southwest. This may be due to the rise of greening in Guangdong Province in 1985. The forest area in low-altitude areas increased, and the distribution of carbon storage tended to

be uniform. From 1993 to 1996, China entered a period of rapid economic development, the phenomenon of deforestation in low altitude areas increased with urbanization, and the distribution of carbon storage changed again. Since 1996, China vigorously developed ecological construction and paid attention to forestry development [56]. In economically developed areas with low-altitude, afforestation efforts are strong, which reduces the gap with high-altitude and economically backward areas, and the forest carbon storage gradually presents a uniform distribution state. Compared with northern Guangdong, the Pearl River Delta was rapid economic development and less forest area. Therefore, although the center of gravity of forest carbon storage in Guangdong Province shifts from 1997 to 2012, it is generally close to northern of Guangdong. It can be seen from the above analysis that the implementation of forestry policies, such as returning farmland to forest, ecological compensation for public welfare forests, and reform of collective forest tenure system, has made great contribution to the steady increase of carbon storage. The dynamic changes of forest carbon storage are greatly affected by forestry policies. Secondly, economic development and human disturbance also have an impact on the dynamics of the forest carbon storage.

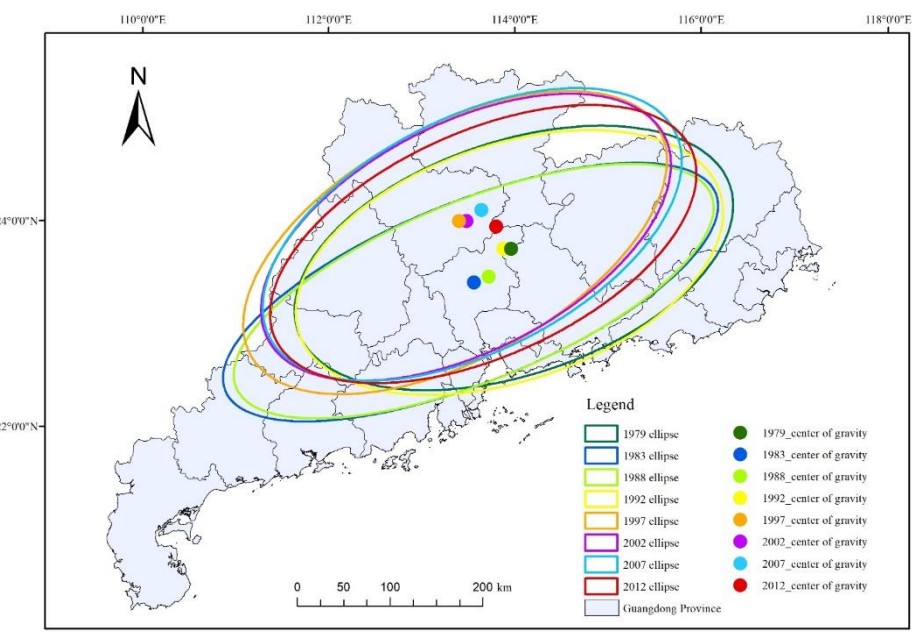

**Figure 5.** Spatial changes of forest carbon storage in Guangdong Province from 1979 to 2012.

### 3.2. Driving Factors for Forest Carbon Density

From three aspects of environmental factors, understory factors, and forest stand factors, forest carbon density data in 2012 were chosen as a representative to analyzes interactions among them and their impacts on forest carbon density. Eight factors including slope, aspect, position, altitude, landform, humus thickness, soil thickness, and soil texture were assumed to reflect environmental factors. Six factors, including shrub height, shrub coverage, herb height, herb coverage, total vegetation coverage, and litter thickness, were selected to reflect understory factors. Five factors, such as average age, average DBH, average height, canopy density, and dominant tree species, were chosen to stand for stand factors. In this paper, the optimization of driving factors is carried out by the statistical test method, that is, correlation analysis. Correlation analysis refers to the analysis of two or more related variable elements, so as to measure the degree of correlation between the two variables [57]. Since forest carbon density in Guangdong Province does not follow normal distribution, in this paper, SPSS was used to do Spearman correlation analysis, and the driving factors with significant correlation were selected and added into the SEM. In SPSS 26.0 [58], the variance inflation factor (VIF) test method is used to delete variables that have multiple commonalities (that is, VIF > 10). After two screenings, the following

15 forest carbon density driving factors in Table 4 are finally obtained. The construction and verification of the SEM (SEM) are carried out using AMOS 22.0.

**Table 4.** The correlation between driving factor and carbon density.

| Driving Factor | Correlation | Driving Factor | Correlation |
|---|---|---|---|
| Slo | 0.561 ** | Shr_H | −0.462 ** |
| Slo_P | −0.432 ** | Her_H | −0.205 ** |
| Slo_A | −0.399 ** | Ave_A | 0.884 ** |
| Alt | 0.450 ** | Ave_D | 0.921 ** |
| Soi_T | 0.504 ** | Ave_H | 0.911 ** |
| Hum_T | 0.604 ** | Cro_D | 0.902 ** |
| Lit_T | 0.642 ** | Dom_T_S | 0.617 ** |
| Shr_C | 0.456 ** | | |

Note: ** is significant at the 0.01 level. Slo is slope, Slo_P is slope position, Slo_A is slope aspect, Alt is altitude, Soi_T is soil thickness, Hum_T is humus thickness, Lit_T is litter thickness, Shr_C is shrub coverage, Shr_H is shrub height, Her_H is herb height, Ave_A is average age, Ave_D is average DBH, Ave_H is average height, Cro_D is crown density, Dom_T_S is dominant tree species.

The 15 selected driving factors were added to the model of SEM, after repeated testing, only 13 driving factors are retained, and finally the optimal SEM is built (Figure 6). $\chi^2/df$ is 1.901, GFI is 0.955, and RMSEA is 0.056, indicating that the SEM constructed is basically ideal. As shown in Table 5, $p$ showing *** means that the two latent variables are significantly correlated at the level of 0.001. In the process of model testing, since the forest stand factor is less significant than the understory factors, this relationship is deleted and the relationship of the forest stand factor and the environment factor as well as the relationship of the understory factors and the environment factor is retained.

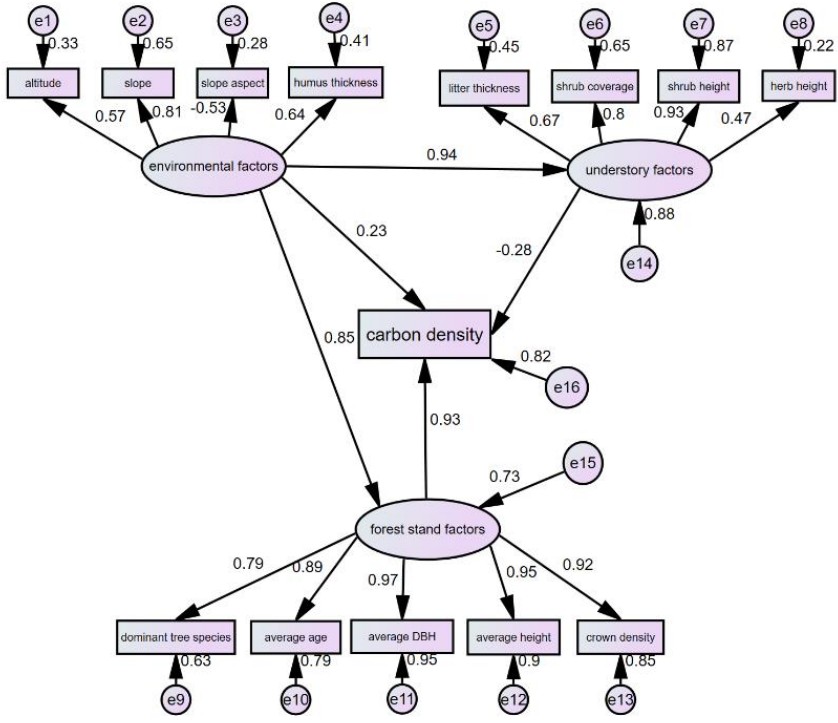

**Figure 6.** Structural equation model of carbon density in Guangdong Province.

**Table 5.** Latent variable correlation.

|  | Estimate of Regression Weight | S.E. | C.R. | *p* |
|---|---|---|---|---|
| forest stand factor < environment factor | 0.056 | 0.005 | 8.957 | *** |
| understory factors < environment factor | 0.011 | 0.001 | 7.802 | *** |

*p* showing *** means that the two latent variables are significantly correlated at the level of 0.001.

As illustrated in Figure 6, it can be seen that forest stand factors have the greatest impacts on carbon density (0.93), the second is the understory factors (−0.28), which are negatively correlated with carbon density, and the influence of environmental factors is the least (0.23), while environmental factors have a significant positive impact on stand factors and understory factors, and the path coefficients are 0.83 and 0.94, respectively. The SEM showed that all driving factors reached significant levels. The load coefficients of standardized factors in environmental factors from large to small are slope (0.81), humus thickness (0.64), altitude (0.57), and slope aspect (−0.53). Among stand factors, the highest load factor of standardized factor is the average age (0.99), followed by the average DBH (0.97), the average height (0.95), the canopy density (0.92), and the lowest is dominant tree species (0.72). Among the understory factors, the load coefficients of standardized factors from large to small are the average height of shrub (0.93), shrub coverage (0.80), thickness of litter (0.67), and average vegetation (0.47).

The influence of forest stand factors on the forest carbon effect is very significant, understory factors and environmental factors have certain influence on forest carbon, forest stand and environmental factors and forest carbon present positive correlation, understory and forest carbon negative correlation, and environmental factors can indirectly affect forest carbon density through influencing forest understory and forest stand. Forest stand factors have an extremely significant impact on forest carbon density. This is because the average DBH and average tree height of the forest directly determine the stock volume of the forest and thus affects the carbon density of the forest. Environmental factors have a significant correlation with forest stand factors, indicating that the slope, altitude and humus layer of environmental factors directly affect the absorption of light, heat and nutrients by the forest. There is a negative correlation between understory factors and forest carbon density. This may be understory plants compete with arbor trees for nutrition and living space, resulting in a slower increase in forest carbon density when understory plants are flourishing.

## 4. Discussion

Based on the NFCI from 1979 to 2012, the biomass conversion factor method is used to calculate the long-term carbon storage and carbon density in Guangdong Province. The analysis of temporal and spatial dynamics and driving factors of forest carbon storage and carbon density can objectively evaluate the long-term effects of forestry policies, human economic activities and urbanization on the function of forest carbon sink, so as to provide a certain scientific basis for the making of long-term sustainable management planning at the provincial level.

The total forest carbon storage in Guangdong Province increased from $4372.91 \times 10^4$ t in 1979 to $19,460.84 \times 10^4$ t in 2012, with an average annual increase of $457.21 \times 10^4$ t; carbon density increased from 12.28 t/ha to 26.77 t/ha, an average annual increase of 0.44 t/ha. From 1979 to 2012, the proportion of carbon storage of broadleaf forest gradually increased, and by 2012, the proportion had increased to 58.89%. Although the proportion of carbon storage in young and middle-aged forest shows a downward trend, it still occupies a dominant position. The slope univariate linear regression analysis showed that the forest carbon storage of Guangdong Province presented an overall gradual upward trend. The standard deviation ellipse analysis showed that the spatial distribution of forest carbon storage had become gradually uniform, and the center of gravity shifted to the northern Guangdong. The results of SEM showed that forest carbon storage was significantly

positively correlated with forest stand factors such as average age, average DBH, average height, and canopy density, and negatively correlated with understory factors.

In this study, the forest carbon storage and carbon density in Guangdong Province show an overall upward trend, which is the same as the change trend of the national forest carbon storage during the same period. In 2007, the forest carbon density of Guangdong Province was 23.02 t/ha, which was very close to the research results of other scholars (23.11 t/ha) [59]. Compared with the forest carbon density in other countries in the same period, the forest carbon density of eastern Asia was 34.40 t/ha, that of Oceania was 54.80 t/ha, that of central America was 90.40 t/ha, that of north America was 55.00 t/ha, and that of west-central Africa was 116.90 t/ha [60,61]. Compared with carbon density in national average and other provinces in China, the national average forest carbon density was 38.05 t/ha [62], the carbon density in Fujian Province was 65.65 t/ha [63], in Yunnan Province it was 50.58 t/ha [64], in Jiangxi Province it was 25.38 t/ha [65], and in Hunan Province it was 18.53 t/ha [66]. The carbon density in Guangdong Province is significantly lower. The huge difference of forest carbon density between Guangdong and other regions is mainly related to forest type, forest origin, and management intensity in different regions. The proportion of natural forests in Oceania and central America is dominant, while artificial secondary forests are the majority in Guangdong Province, which are mainly young and middle-aged forest. At the same time, the area percent of rural collective owned and farmer owned forest in Guangdong is over 90%. Due to the long rotation and the low rate of economic return, farmers are not willing to actively manage the forest, resulting in low forest management intensity and low stock volume per unit area. Compared with other provinces at the same latitude range, the forest carbon density in Guangdong Province is lower and significantly lower than the national average, mainly due to the large number of young and middle-aged forest and small volume per unit area. Therefore, we should actively adjust the age group structure, actively carry out forest tending increase the proportion of over mature forests, prolong the rotation period of trees, and select native broadleaf trees with long life span and highly efficient carbon sequestration ability for planting and cultivation. At the same time, plant mixed forest to improve the full cycle coverage of forests, properly control the tree density reasonably improve the stand quality, and enable trees to obtain sufficient light and nutrients to improve photosynthesis efficiency and reduce carbon emissions. The growth of understory vegetation should be controlled reasonably, the thickness of litter should be maintained, and the carbon storage capacity of forest should be brought into full play. The temporal and spatial dynamics of forest carbon storage and carbon density in Guangdong Province are greatly influenced by forestry policies and socio-economic conditions. Therefore, forestry policies such as returning farmland to forest, building beautiful villages and compensating for ecological benefits of public welfare forests should be strengthened to reduce the negative impact of social economic development and urbanization process on forest carbon sink.

At this stage, there is still a big gap in the results of forest carbon storage calculated by different methods in the same region. In this study, since the biomass expansion factor equation of each tree species provided by Guangdong Forest Resources Monitoring Center does not include all tree species in the study area, the national biomass expansion factor equation is used to supplement, and there may be errors in the results. However, compared with the research results of other scholars, the results are very close, so the estimation results can be adopted. In the future, different research methods will be used to estimate carbon storage in Guangdong Province, so as to find the most accurate estimation method and provide a more accurate scientific basis for accounting provincial carbon sequestration.

**Author Contributions:** Conceptualization: T.L. and M.-Y.L.; methodology: T.L. and M.-Y.L.; software: T.L.; validation: T.L. and L.T.; formal analysis: T.L.; resources: M.-Y.L.; data curation: T.L. and L.T.; writing—original draft: T.L.; writing—review and editing: T.L., M.-Y.L. and L.T.; visualization: T.L.; supervision: M.-Y.L.; project administration: M.-Y.L.; funding acquisition: M.-Y.L. All authors have read and agreed to the published version of the manuscript.

**Funding:** This research was financed by the National Natural Science Foundation of China (No. 31770679). We would like to thank the Top-notch Academic Programs Project of Jiangsu Higher Education Institutions, China (TAPP, PPZY2015A062), for its linguistic assistance during the preparation of this manuscript.

**Conflicts of Interest:** The authors declare no conflict of interest.

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
