# Peer review of "Dynamics of Carbon Storage and Its Drivers in Guangdong Province from 1979 to 2012"

_forests, doi:10.3390/f12111482_

Round 1

Reviewer 1 Report

Dear Authors!

The content of the paper is very valuable because of the long timeline dataset. However, the significance of the manuscript and discussion is very interesting, there are some confusing parts for me. Please see my suggestions, questions, and comments below.

The abstract summarized the essence of the research but I suggest inserting a sentence about the objectives. I did not find it well written in text (even later).

About the introduction part, line 31-33 - Please insert a citation. 

It would greatly strengthen the manuscript to cite papers and examples from the rest of the world not only from the author's country. It would be nice to get information about other long-term datasets or carbon inventories not only "in China and abroad" approach. 

Based on the checking of the references, there are only 12 citations for foreign authors from the 48 cited references. Please try to find balance.

Please, clarify and highlight the main objectives of the research at the end of the introduction; therefore, it is easier to summarize the realized results. Why do you choose the study area? 

In research methods, the paper presents the structural equation model. What about the reliability of the data? How much uncertainty is there in the model? What are the weaknesses? Please insert some sentences about the evaluation of the model.

Figure 2 - it is hard to understand. What is 100%? Please clarify the term. Furthermore, it is suggested to clarify the terms overmature or mature or near-mature forest. The paper analyzed the carbon stock of a lot of tree species (coniferous, bamboo, etc.). As far as I know, these mentioned species do not exceed the (over-, near)maturing level at the same age.

English is fine, there are only a few spelling errors in the text.

Reviewer 2 Report

Review of forests-1393557

General Remarks

This paper can be greatly improved and, if so, would be a very high-quality paper. Harvesting must be taken into an account; it will greatly explain shrub layer and shrub height. Carbon density needs to be compared to estimates in literature of small, using all forest types throughout the world to give this more than just the forest in Guangdong. Add “and comparisons to other forest types in the world” to the end of the current title.  It will get many more readers. These additions (and remarks below) will require a change in the abstract.

Specific Remarks.

Line 8; change grasping to deriving

Line 24; I don not understand what the ‘The forest structure is gradually rationalized” means.

Line 26; by the way, destructive sampling is not extremely destructive to ecosystems,  Stands are chosen, and  individual tree are sampled in the stand. Then you take derived allometric and use to estimate stand biomass.

Line 52;  what did YOU use to convert biomass to biomass to carbon? Carbon content is generally about 0.50.

Line  74; “human activities” does include   urbanization, but it also includes harvesting. See general remarks above.

Line 28; Need comparisons to the rest of the world.  Find ecosystems like yours in the literature.  Include  estimates in other stands  calculated in all methods.

Line 85;  deviation ellipse; This need to be described in a  (  ) after the term.  All  other terms like this and abbreviations need to have this the first time they are used in the paper

 Lines 151-154; you use W1 – W4 in equations I -4. But you use all W1 in descriptions.  I suggest  using

 Wstem, Wbranch, Wleaves,  Wroots,

Line 256; What volume equation did you used  for bamboo.  I can’t be the same as for trees.

Line 262; I don’t understand what “cash” means in cash forest.

Table 2; I don’t know what hm-2means.  Please educate me.

Line 396-397; You need to provide a citation or two for this statement.  I can’t just be said without backing it up.

Round 2

Reviewer 2 Report

Review of re-submitted manuscript - forests-1393557

Still can be greatly improved:

I have added some comments on different lines.

You need to detail what allometry you use for roots. Add Total C, above ground C, and Coarse root C to Table 1 and add root:shoot ratio. It would be very interesting to compare R:S among forest types.  Potential interpretable biological information that can inform

Point 2. “reasonable is not an ecological term. Say exactly what you mean using ecological terms.

Point 4. I insist of this! A reader should not have to go to another paper to get the carbon contents you use. List all YOU!!!! used. 0.4, 0.5 or whatever for each species that you used.  You could make a simple table and cite the literature in the table heading.

Point 5. I still think you should add “years since harvest” to your variable and it might  help explain shrub layer.

Point 6. You need to put more of your incites to comparisons with different estimates across the globe.  Why do they differ?  Can you suggest ways that different methodologies can be combined to make estimates across the globe or a continent? You need to also consider estimates via Eddy Flux where above and below biomass estimates are used.

Point 10.  Cash is still not a good word.  You might use forest where economically important products are harvested.

Line 26. Add after line 26, we also compare and consider estimates made across the globe in similar forest types.

Line 110.  Add 4th; compare and consider estimates of similar forest across the globe.

Line 349.  Obviously, I don’t like “cash”.  See above.

Line 617. Make “gradual upward trend”.

Add at the end of the title 'and comparisons to estimates of similar forests across the globe".
